# Extraction of Offshore Aquaculture Areas from Medium-Resolution Remote Sensing Images Based on Deep Learning

**Yimin Lu [1,*], Wei Shao [1] and Jie Sun [2]**

1   Key Laboratory of Spatial Data Mining & Information Sharing of Ministry of Education,
    National Engineering Research Centre of Geospatial Information Technology, Academy of Digital
    China (Fujian), Fuzhou University, Fuzhou 350116, China; n195520015@fzu.edu.cn
2   School of Geography and Information Engineering, China University of Geosciences, Wuhan 430074, China;
    jsun20@gmu.edu
*   Correspondence: luym@fzu.edu.cn; Tel.: +86-138-5015-9899

**Abstract:** It is important for aquaculture monitoring, scientific planning, and management to extract offshore aquaculture areas from medium-resolution remote sensing images. However, in medium-resolution images, the spectral characteristics of offshore aquaculture areas are complex, and the offshore land and seawater seriously interfere with the extraction of offshore aquaculture areas. On the other hand, in medium-resolution images, due to the relatively low image resolution, the boundaries between breeding areas are relatively fuzzy and are more likely to 'adhere' to each other. An improved U-Net model, including, in particular, an atrous spatial pyramid pooling (ASPP) structure and an up-sampling structure, is proposed for offshore aquaculture area extraction in this paper. The improved ASPP structure and up-sampling structure can better mine semantic information and location information, overcome the interference of other information in the image, and reduce 'adhesion'. Based on the northeast coast of Fujian Province Sentinel-2 Multispectral Scan Imaging (MSI) image data, the offshore aquaculture area extraction was studied. Based on the improved U-Net model, the $F1$ score and Mean Intersection over Union (MIoU) of the classification results were 83.75% and 73.75%, respectively. The results show that, compared with several common classification methods, the improved U-Net model has a better performance. This also shows that the improved U-Net model can significantly overcome the interference of irrelevant information, identify aquaculture areas, and significantly reduce edge adhesion of aquaculture areas.

**Keywords:** medium-resolution remote sensing image; offshore aquaculture area; deep learning; U-Net; classification

## 1. Introduction

The vigorous development of China's offshore aquaculture industry [1] has provided tremendous help to China's economic development while also providing necessary nutrients for the growing population [2]. In 2019, the total amount of marine aquaculture in China was 20.65 million tons, accounting for 40.7% of the total aquatic product cultivation, a year-on-year increase of 1.76% [3]. The commonly used methods of offshore aquaculture are floating raft aquaculture and cage aquaculture, which are usually located in coastal waters [4]. However, due to the lack of reasonable management and control, excessive aquaculture, and other factors, the development of offshore aquaculture has also brought a series of problems, such as the eutrophication of the aquaculture sea area [5], bringing a negative impact on the sustainable development of the marine ecosystem [6].

Remote sensing technology can overcome the shortcomings of traditional field surveys and realize full-time and large-scale monitoring [7]. It is an effective means to achieve dynamic monitoring of offshore aquaculture and has been widely used. In recent years,

researchers have proposed many classification methods for aquaculture areas based on remote sensing images.

Sridhar et al. constructed a saltpan index based on the VIS (Green) and short-wavelength infrared (SWIR) bands and extracted the confusing saltpans and aquaculture area from remote sensing images [8]. Lu et al. constructed an aquaculture area index based on the different spectral characteristics of different types of aquaculture areas on remote sensing images and combined texture information with the shape characteristics of the aquaculture area to achieve the classification of aquaculture areas [9]. Tang et al. combined the spectral information and texture feature information of remote sensing images and used decision trees as classifiers to extract aquaculture areas from remote sensing images [10]. Chu et al. used the support vector machine (SVM) algorithm to successfully extract the floating raft aquaculture area from the high-resolution satellite image of Gaofen-1 (GF-1) based on the spectral information and texture feature information of the remote sensing image [11]. Most of the above methods are classified by constructing a spectral index combined with traditional machine learning algorithms. However, the construction of a spectral index and the feature selection of traditional machine learning algorithms often require the support of relevant professional knowledge and experience. On the other hand, the above method has high precision in small-scale offshore aquaculture extraction, but it is difficult to maintain a good extraction effect in large-scale offshore aquaculture extraction [12].

The accuracy of the results based on machine learning algorithms is more likely to encounter bottlenecks. With the great success of deep learning in the field of computer vision, remote sensing scholars increasingly apply deep learning to the field of semantic segmentation of remote sensing images [13,14], and aquaculture based on remote sensing images extraction is also one of them. Compared with traditional extraction methods for offshore aquaculture areas, deep learning methods are used to extract offshore aquaculture areas, which avoids the need to perform preprocessing operations such as water and land separation on remote sensing images first and also improves the extraction efficiency. At the same time, the deep learning method has a better analysis ability in the face of the dense distribution and complex spectral information of the offshore aquaculture area [15] and compared with the traditional algorithm (random forest, support vector machine, etc.), it can extract more abstract and useful features [16] and has better extraction accuracy.

Cui et al. used a method based on FCN (Full Convolutional Neural Network) [17] by adding L2 regularization and dropout strategies to avoid overfitting and extracted the offshore floating raft culture area in Lianyungang [18]. Fu et al. proposed a hierarchical cascade structure composed of atrous convolution to obtain contextual information while using the attention mechanism to optimize the feature matching of different levels of feature maps to better identify offshore aquaculture areas of various sizes [19]. Sui et al. adjusted the image display method based on the spectral characteristics of the offshore aquaculture area and at the same time used GAN (Generative Adversarial Networks) [20] to generate training data to make up for the lack of training data. The above strategy improved the extraction accuracy of the offshore cage and floating raft aquaculture area [21].

The U-Net model is a commonly used model in the field of semantic segmentation. Compared with other models in semantic segmentation, the U-Net model has smaller parameters and is convenient for training and prediction [22]. It was initially applied to biomedical tasks and then gradually applied to the classification of remote sensing images. Yang et al. used the U-Net model to classify ground land cover types and achieved better results compared with other methods [23]. In addition, the U-Net model requires repeated down-sampling of images, which will inevitably lose location information, leading to smooth edges of objects [24]. In view of the above problems, many researchers have improved the U-Net model. Cui et al. improved the decoder part of the U-Net model and proposed a PSE structure based on an SPP structure for the decoder part. The improved model can help capture the edge information of aquaculture areas in feature maps of different sizes and effectively reduce the problem of "adhesion" between adjacent aquaculture

areas [25]. Cheng et al. also improved the U-Net model using Hybrid Dilated Convolution (HDC) [26] to expand the network receptive field, thus as to obtain more semantic information [27]. The above research mainly focuses on one kind of aquaculture area, such as floating raft areas. Secondly, the above method mainly aims to further improve the information extraction ability of U-Net model without considering further utilization of extracted information.

In recent years, researchers investigating the extraction of offshore aquaculture areas based on deep learning methods have mostly focused on high-spatial-resolution remote sensing images [28]. However, the use of high-resolution remote sensing images to extract large-scale offshore aquaculture areas requires massive amounts of data and computing resources. At the same time, high-resolution remote sensing images are usually not available for free [29]. Therefore, the extraction of large-scale offshore aquaculture areas based on high-spatial-resolution remote sensing images is difficult to achieve. The marginal information between different aquaculture areas in low-resolution remote sensing images is indistinguishable, thus it is difficult to accurately extract offshore aquaculture areas based on low-resolution remote sensing images. Medium-resolution remote sensing images are undoubtedly a better choice, but the borders of offshore aquaculture areas in medium-resolution remote sensing images are more blurred than high-resolution remote sensing images and are more susceptible to the influence of coastal land and seawater, and spectral information is more complicated, which leads to the difficulty of offshore aquaculture area extraction. Therefore, we propose an improved U-Net model for medium-resolution aquaculture area extraction. The motivations behind the development of the model include two aspects. Firstly, we hoped that the model could better mine the relevant information of aquaculture areas from medium resolution remote sensing. The improved U-Net model uses the improved ASPP structure to mine semantic and location information of aquaculture areas from feature maps at different scales and uses the flow alignment module (FAM) [30] to replace the traditional up-sampling method to better match feature maps at different levels thus as to make full use of the relevant information of aquaculture areas. Secondly, both the ASPP (atrous spatial pyramid pooling) structure and FAM inevitably retain some irrelevant information when acquiring relevant information. The improved U-Net model uses an attention mechanism to filter information. At the same time, the attention mechanism also makes the model pay more attention to the information related to the aquaculture area and further overcome the interference of irrelevant information.

The rest of the paper is structured as follows: the Section 2 mainly introduces the research area of the experiment, the relevant data of the experiment, and the processing method. The Section 3 introduces the improved U-Net model proposed in detail. Finally, the Sections 4 and 5 introduce the details of the experiment and provide a discussion and conclusion.

## 2. Study Area

The study areas selected in this paper were Sansha Bay and Luoyuan Bay in the northeastern sea area of Fujian Province, and their geographic scope is between $26°10'\sim26°48'$N and $119°34'\sim120°09'$E. The sea area has superior natural resources and rich marine fishery resources, which provides very favorable conditions for the development of offshore aquaculture. There are mainly two types of aquaculture areas in the experimental area: the floating raft aquaculture area and the cage aquaculture area. The floating raft aquaculture area is shown as a dark rectangular strip on the image, which is mainly used for cultivating kelp, seaweed, and mussels; the cage aquaculture area is gray-white on the image, and it is mainly used for fish and shrimp [27], as shown in Figure 1.

The data used in this study were Sentinel-2 Multispectral Scan Imaging (MSI) image data; the shooting date was 13 February 2018; the spatial resolution was 10 m, and it contained four bands (red band, green band, blue band, and near-infrared band); the pixel size was 7095 × 7030, covering the sea area where Sansha Bay and Luoyuan Bay are located. First, we performed atmospheric correction and other preprocessing operations on remote

sensing images, and then marked the processed images, which were divided into three categories: background, raft aquaculture area, and cage aquaculture area, with 0 for the background information and 1 for the cage aquaculture zone; 2 represents the floating raft aquaculture zone. In order to avoid memory overflow, this experiment was based on the sliding window method to segment the remote sensing image. The image block with a pixel size of 512 × 512 was selected as the segmentation window, and the sliding step was 256.

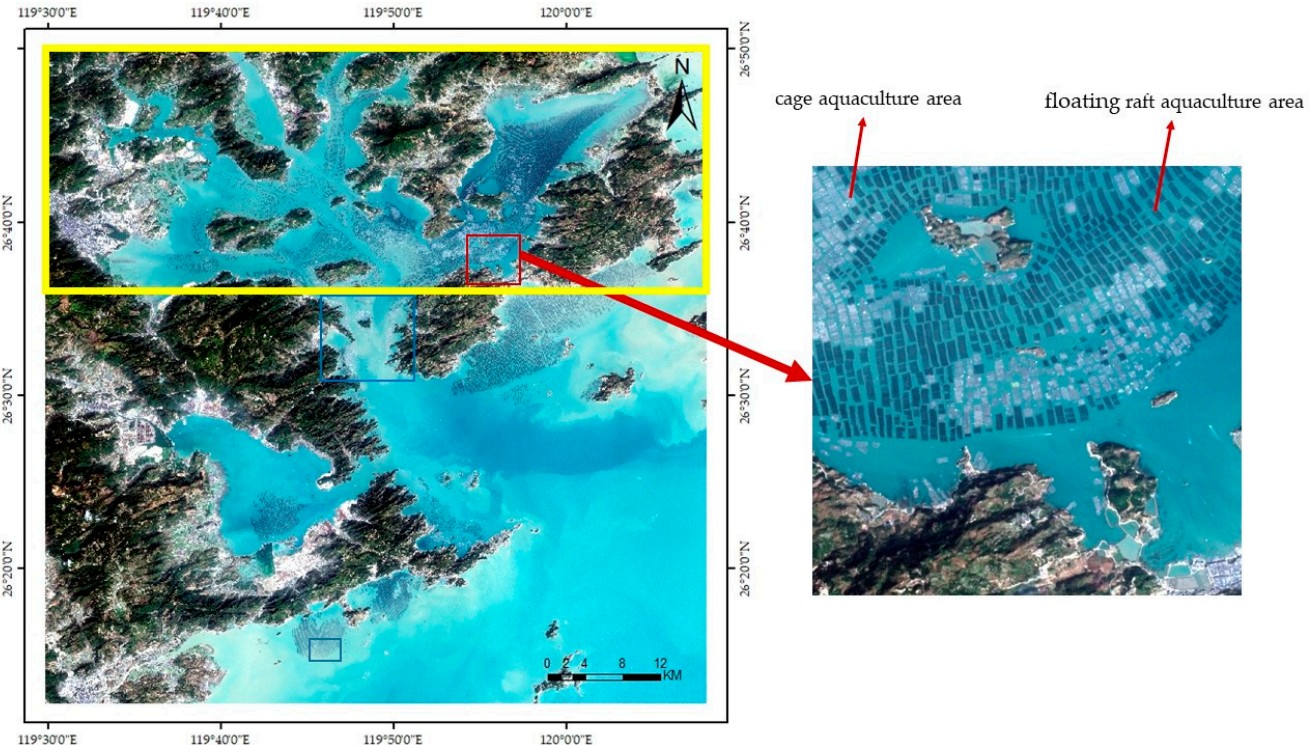

**Figure 1.** The sea area where Sansha Bay and Luoyuan Bay are located is an important marine aquaculture area in Fujian Province. The left of the picture here is the geographic location of the study area, and the right of the picture is an example of images of the cage aquaculture area and the floating raft aquaculture area. The yellow box area of the image was mainly used for training and validation, while other areas were used for testing. The blue box area of the image was the typical study area we chose.

## 3. Methods

In this section, we first introduce the overall structure of the improved U-Net model. The improved U-Net model is mainly composed of three parts. Specifically, the traditional convolutional neural network was first used as the feature extraction network, and the corresponding feature map was obtained according to the input remote sensing image. The encoder part obtained feature maps of different scales, and these feature maps contained semantic information and location information of the offshore aquaculture area. After that, the feature map was input into the decoder by the connection layer, and according to the feature maps of different levels, it was gradually restored to the input image size. Next, we will describe the details of the improved U-Net model, including the improved ASPP structure and up-sampling structure.

### 3.1. The Proposed Improved U-Net Model

In this work, the U-Net model was selected as the backbone of the network. The U-Net model [22] consisted of an encoder part and a decoder part, and the encoder and decoder were connected through an intermediate layer. The encoder part can extract image features, and the decoder part was used to gradually restore the feature map obtained by the encoder to the original image size. Therefore, the feature extraction capability of the

encoder part and the image restoration capability of the decoder part directly affect the performance of the entire network.

The improved U-Net model proposed in this paper used Efficientnet [31] as the feature extraction network in the encoder part. Efficientnet has better accuracy and a faster speed than traditional networks such as ResNet [32], DenseNet [33], and Xception [34]. An improved ASPP structure was used between the encoder and the decoder. The decoder part restored the feature map size through the proposed up-sampling structure, which was composed of a flow alignment module and an ECA (Efficient Channel Attention) module [35]. Additionally, the encoder and decoder parts of the same level were connected through skip connection, which contained an attention module composed of strip pooling [36]. The overall structure of the improved U-Net model is shown in Figure 2.

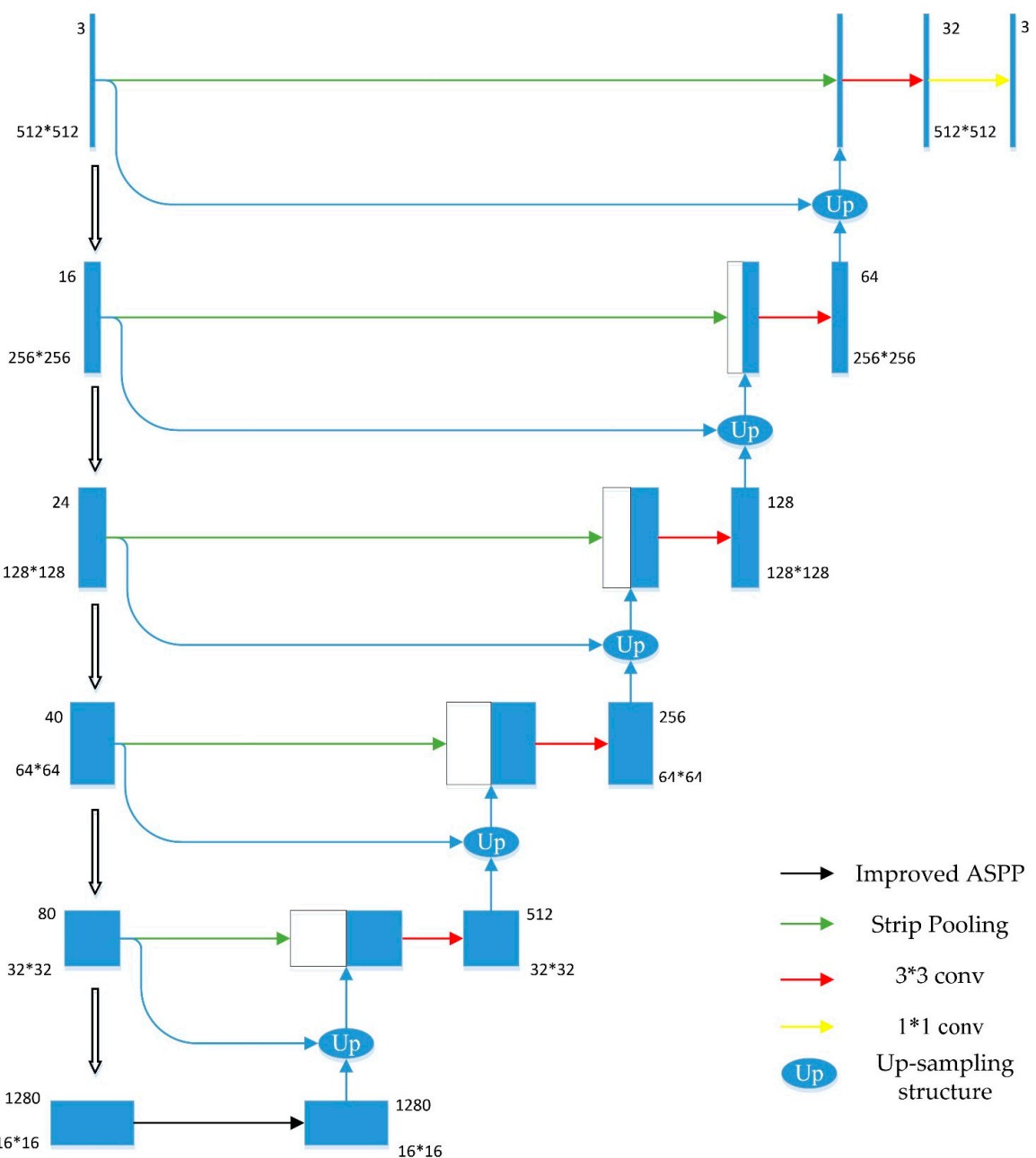

**Figure 2.** The improved U-Net model. Each blue rectangle represents a multi-channel feature map, and the white rectangle represents the feature map obtained by bar pooling. The number of channels is at the top of the rectangle, and the size is at the bottom.

### 3.2. The Improved ASPP Structure

Context information helps to distinguish the target object from the background information [37]. Therefore, the ability to capture multi-scale information is of great significance in solving the problem of offshore aquaculture areas easily being disturbed by background information in medium-resolution remote sensing images.

In order to obtain contextual information, we chose the ASPP structure as the main body for research. ASPP (atrous spatial pyramid pooling) structure [38] realizes pyramid-shaped void pooling on the spatial scale. The traditional ASPP structure is usually composed of a $1 \times 1$ convolution, three $3 \times 3$ atrous convolutions with different sampling rates, and a spatial pooling. In this structure, feature maps of different scales can be obtained by setting different sampling rates.

In order to obtain contextual information more effectively and make it more suitable for extraction tasks in offshore aquaculture areas, the traditional ASPP structure was improved. The improved ASPP structure proposed in this paper, as shown in Figure 3a, mainly uses strip pooling instead of traditional spatial pooling. As the sampling window of the traditional spatial pool is square, when the target object is a long strip, such as an offshore aquaculture area, the square window will inevitably contain interference information from other unrelated areas, while for the strip pool, long strip sampling windows, sampling reduces the acquisition of irrelevant information, and reduces the interference of the above problems to a certain extent [36]. The improved ASPP structure is shown in Figure 3b.

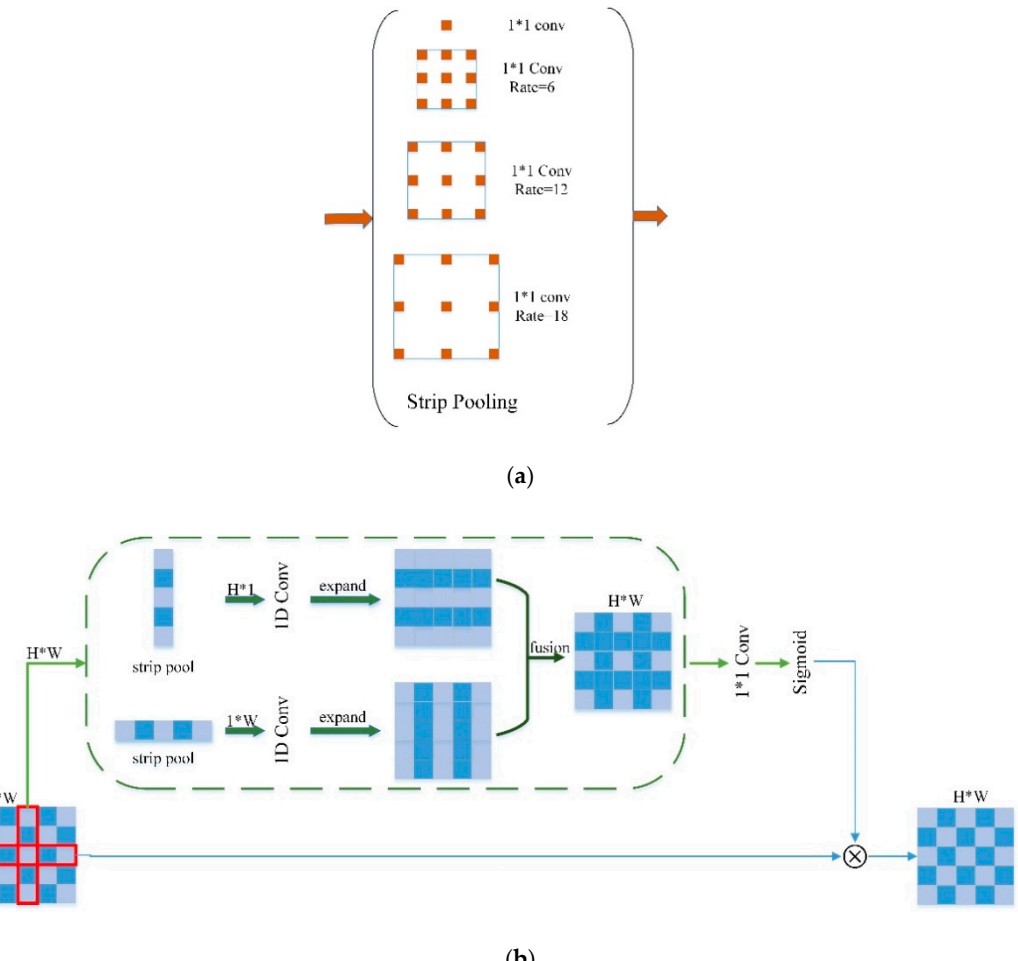

(a)

(b)

**Figure 3.** The improved ASPP structure, composed of atrous convolution and strip pooling. (**a**) The overall structure of the improved ASPP structure. (**b**) Strip pooling structure diagram.

### 3.3. The Up-Sampling Structure

The encoder–decoder structure network generally transfers the feature map obtained by the encoder part to the decoder part through a skip connection to gradually restore the original image size. The decoder partly restores the image size, usually using interpolation methods, but the interpolation method does not combine the information of low-level features and high-level features very well, and it is very prone to misalignment of semantic information. Li et al. proposed a flow alignment module (FAM). By referring to the optical flow idea in the field of video semantic segmentation, different levels of features were used to construct the offset field, and then the image size was gradually restored through the offset field [37].

This method first processes the feature maps of different scales and compresses the number of channels of the feature maps to the same. Then, it generates the corresponding offset field according to the feature maps of different scales and uses the generated offset field to restore the image size of the high-level feature to the image size of the low-level feature. Restoring the image size of the feature map through the offset field can make better use of the semantic information of different levels, reduce the problem of semantic misalignment, and further reduce the generation of redundant information. However, this method not only obtains more semantic information but also inevitably obtains more irrelevant information and other interference information, which will affect the classification accuracy of the model.

The Squeeze-and-Excitation (SE) block in squeeze-and-excitation networks (SENet) automatically obtains the importance of each channel by learning the dependencies between different channels, and then pays more attention to the features related to the current task according to the learning results, and suppresses the features that are not related to the current task [39]. Based on this characteristic of the attention mechanism, we considered combining the attention mechanism with the flow alignment module.

However, the Squeeze-and-Excitation (SE) block obtains the importance of each channel by learning the dependencies between all channels. Wang et al. found through research that learning the dependencies between all channels was inefficient and unnecessary. Thus they proposed a more efficient attention module (ECA, Efficient Channel Attention) module [35]. The performance of this module is better than the Squeeze-and-Excitation (SE) block, and at the same time, it hardly increases the complexity of the model. Therefore, we finally chose the ECA module combined with the flow alignment module to form an up-sampling structure.

Specifically, we first used the offset field in the flow alignment module for up-sampling to recover the image size of the high-level feature map. After the up-sampling, we used the residual structure constructed by the ECA module to focus on the information related to the current task and suppress irrelevant information. At the same time, the residual structure can also effectively alleviate the problem of gradient disappearance.

The complete up-sampling structure is shown in Figure 4. First, the low-level feature map and the high-level feature map adjusted the number of channels through a $1 \times 1$ convolution, respectively. Then, we restored the high-level feature map to the size of the low-level feature map, used different levels of feature maps to generate an offset field, and then restored the high-level feature map to the size of the low-level feature map according to the offset field sampling as the output feature map. The output feature map was then input to the residual structure formed by the ECA module, and, finally, the final output feature map was obtained. This structure made better use of the semantic information between different levels while effectively reducing the generation of irrelevant information.

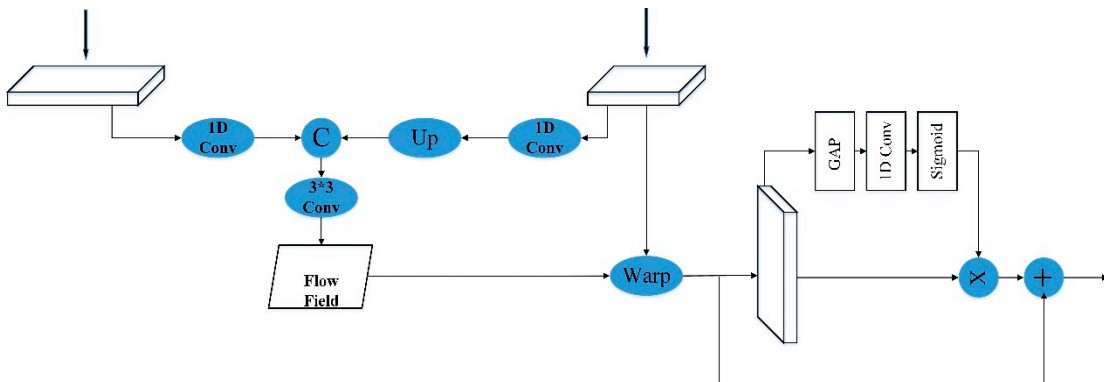

**Figure 4.** The up-sampling structure, composed of a low alignment module (FAM) combined with an attention module. Each black box represents a feature map.

## 4. Results and Discussion

### 4.1. Experiment Setting

The experiment was trained on a server equipped with NVIDIA GeForce RTX 2080Ti 11GB GPU. The experimental model was implemented using the PyTorch framework [40]. The experiment used the Adam optimizer; the initial learning rate was 0.0001; the cosine annealing learning rate strategy was used; the batch size was set to 4; the iteration period was 50 times; the loss function used the cross-entropy function; and the classifier used the softmax function.

In order to avoid overfitting, horizontal, vertical flip, and random noise were used for data enhancement. After data enhancement, there were 1564 images in the training set and 480 images in the validation set, and the image sizes were $512 \times 512$. After the training, a $4100 \times 4000$ remote sensing image was used to verify the effectiveness of the improved U-Net model.

### 4.2. Accuracy Assessment and Comparison

In order to verify the effectiveness of the improved U-Net model, in addition to experimenting with our model, we also added the original U-Net model, DeepLabV3+ model [41], and the traditional machine learning algorithm SVM [42] to extract the floating raft and cage culture area from sentinel-2 MSI image data in the study. To avoid the difference in feature extraction capabilities due to different feature extraction networks, both the original U-Net model and the DeepLabV3+ model in this experiment used EfficientNet as the feature extraction network of the model.

To extract offshore aquaculture areas using traditional machine learning algorithms SVM, first, we used the water index to extract the seawater in the image, and then used the machine learning algorithm SVM to extract floating rafts and cage aquaculture areas from the seawater. We used the more commonly used Normalized Difference Water Index (NDWI) [43]. In the experiment, we used precision, recall, *F*1 score, overall accuracy (OA), kappa coefficient, and Mean Intersection over Union (MIoU) to evaluate the extraction ability of different models for offshore floating raft and cage aquaculture areas. *F*1 score is the harmonic average of precision and recall rate, which can better evaluate the performance of the model. Mean Intersection over Union (MIoU) is a common evaluation index in semantic segmentation. The calculation formulae of the above evaluation index are shown in Formulas (1)–(6):

$$Precision = \frac{TP}{TP + FP} \tag{1}$$

$$Recall = \frac{TP}{TP + FN} \tag{2}$$

$$F1 = 2 \times \frac{precision \times recall}{precision + recall} \tag{3}$$

$$kappa = \frac{p_0 - p_e}{1 - p_e} \tag{4}$$

$$OA = \frac{TP + TN}{TP + TN + FP + FN} \tag{5}$$

$$MIoU = \frac{1}{k+1} \sum_{i=0}^{k} \frac{TP}{TP + FN + FP} \tag{6}$$

where *TP* (true positive), *TN* (true negative), *FP* (false positive), *FN* (false negative), $p_0$, $p_e$ are all calculated according to the confusion matrix, and K represents the number of categories.

### 4.3. Comparison Experiment

First, we analyzed the classification results of each model on the test set from a qualitative perspective, as shown in Figure 5. From the classification results, it can be seen that the classification results based on the traditional machine learning algorithm SVM were significantly worse than the classification results based on several other deep learning methods. A lot of background information was mistakenly divided into floating raft aquaculture and cage aquaculture areas. Compared with the traditional machine learning algorithm SVM, the classification results of the original U-Net model and DeepLabV3+ model significantly reduced the misclassification of floating raft aquaculture area and cage aquaculture area, but there were still some obvious problems of missing points in the cage aquaculture area. The improved U-Net model proposed in this paper can better obtain context information and multi-scale information, and the classification results were significantly better than the machine learning algorithm SVM, the original U-Net model, and the DeepLabV3+ model. The floating raft aquaculture and cage aquaculture areas were almost recognized.

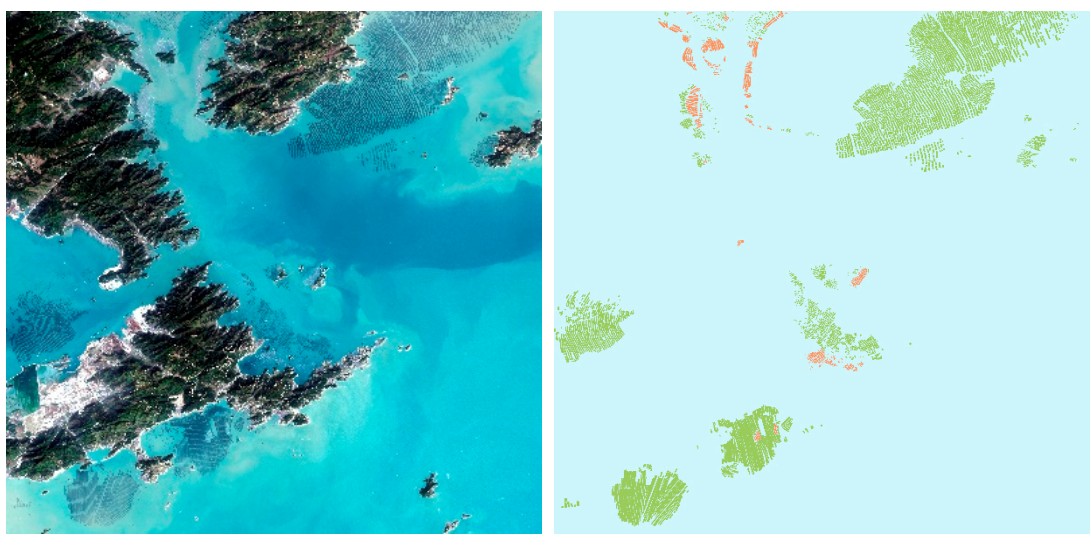

(**a**) Test image.        (**b**) Ground truth.

**Figure 5.** *Cont.*

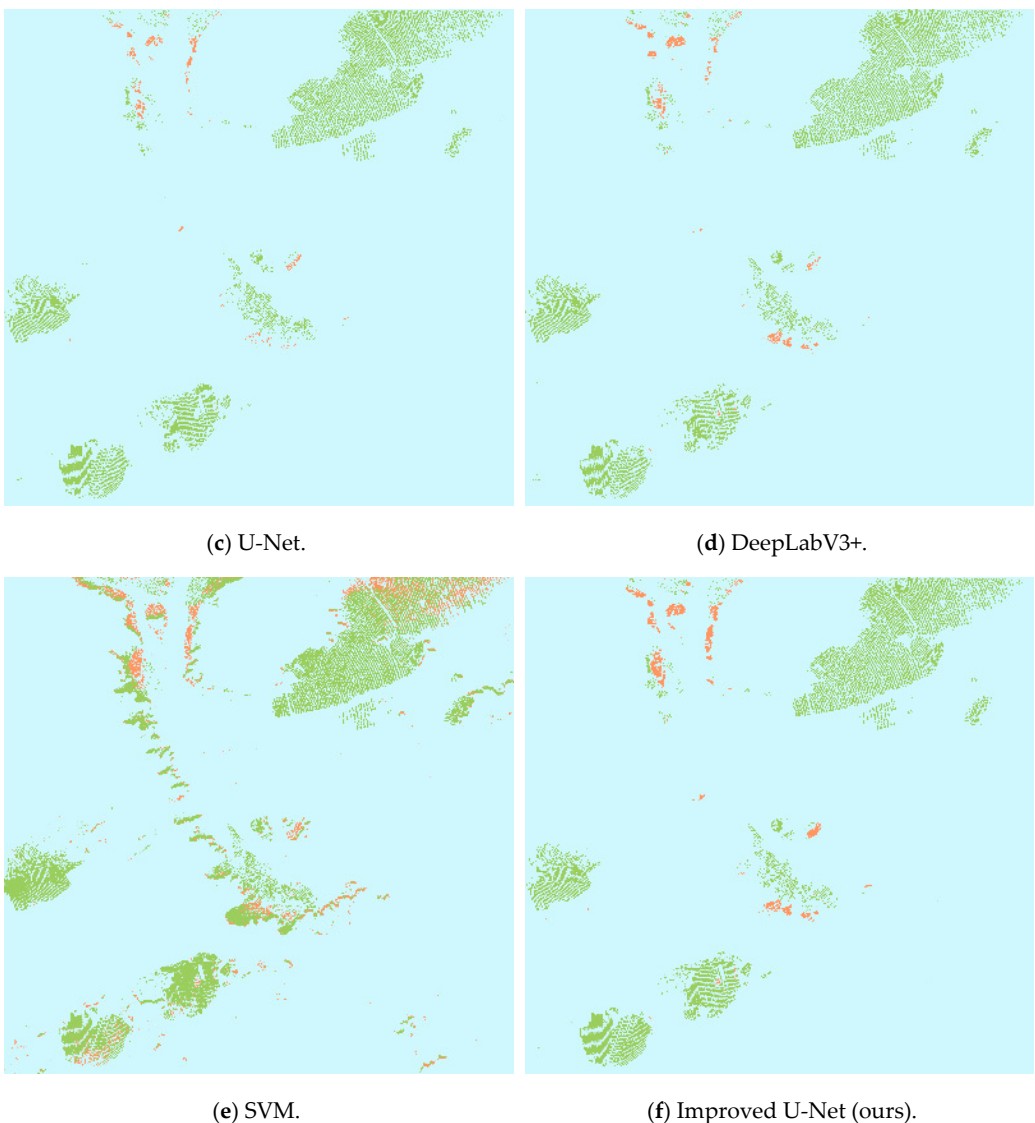

**Figure 5.** The extraction results of the floating raft aquaculture area and the cage aquaculture area on the test image (as the left part of the test area is all land and there is no aquaculture area, this area has been removed): (**a**) test image; (**b**) ground truth; (**c**) original U-Net; (**d**) DeepLabV3+; (**e**) SVM; (**f**) improved U-Net. The orange area in the figure represents the cage aquaculture area; the green area represents the floating raft aquaculture area; and the blue area represents the background.

In order to better evaluate the classification accuracy of each model in the offshore floating raft and cage aquaculture area in the study area, two typical study areas were selected from the classification results to further analyze the model performance. Figure 6 shows the classification results of each model in the typical study area. Figure 6e shows the classification results based on the traditional machine learning algorithm SVM. The machine learning algorithm SVM can be used for floating raft aquaculture and cage aquaculture areas, but many background areas were mistakenly divided into floating raft aquaculture and cage aquaculture areas. Combined with Figure 6a, it can be seen that the misclassified area was often the area with spectral characteristics similar to the floating raft and cage aquaculture area, and the area was often the area at the junction of land and sea. This is mainly because the spectral characteristics of the sea–land junction are more complex, and the traditional machine learning algorithm has a relatively weak feature extraction ability and cannot extract deeper features. Therefore, the complex spectral areas at the sea–land junction were often mistakenly divided into floating raft aquaculture areas or cage aquaculture areas in the classification result map based on SVM.

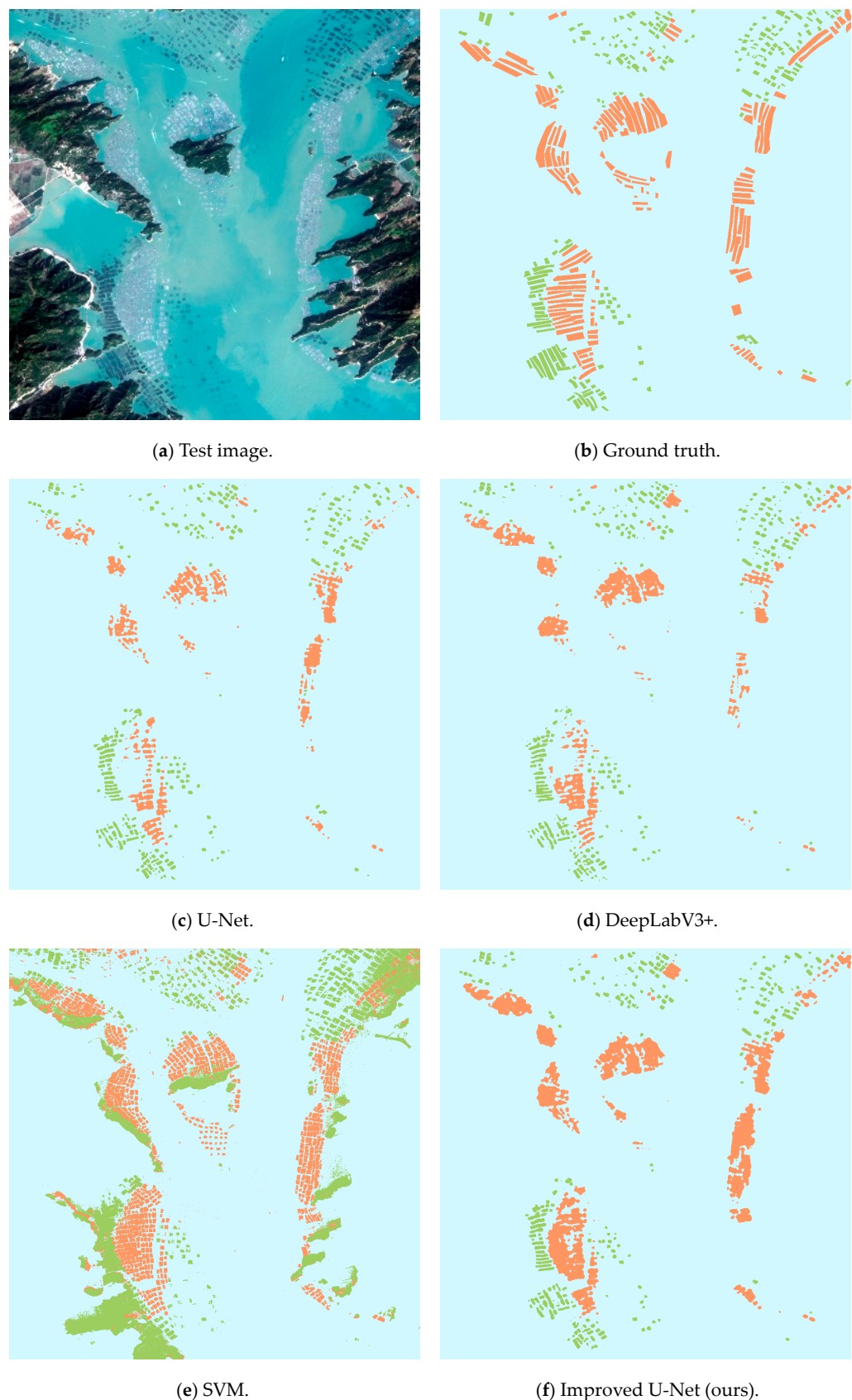

(**a**) Test image.

(**b**) Ground truth.

(**c**) U-Net.

(**d**) DeepLabV3+.

(**e**) SVM.

(**f**) Improved U-Net (ours).

**Figure 6.** Comparison of classification results of different models in typical areas of the study area. The orange area in the figure represents the cage aquaculture area; the green area represents the floating raft aquaculture area; and the blue area represents the background.

The classification results of the original U-Net model and DeepLabV3+ model are shown in Figure 6c and Figure 6d, respectively. From the classification results, it can be seen that these two classification methods have a certain degree of leakage in the extraction of cage aquaculture areas. In combination with the original remote sensing image, it can be seen that this is mainly because the spectral characteristics of this part of the cage aquaculture area are similar to those of the background information, which resulted in this part of the cage aquaculture area being divided into the background. The classification result diagram of the improved U-Net model in this paper is shown in Figure 6f. The floating raft aquaculture area and the cage aquaculture area were recognized, and the cage aquaculture area similar to the background spectral characteristics was not mistakenly classified as the background. The land area with similar spectral characteristics as the floating raft aquaculture and cage aquaculture area was also accurately classified as the background. In summary, the improved U-Net model proposed in this paper has better feature extraction capabilities, can accurately identify floating raft aquaculture areas and cage aquaculture areas, and can avoid misclassification and omission.

In order to further analyze the extraction results of aquaculture areas in each model, we selected a typical region from the prediction results of the model, as shown in Figure 7. In order to simplify the image labeling process, we labeled the aquaculture areas that were relatively close to each other as a whole. Therefore, there may have been some errors between the labeled image and the real image, but such errors can also enhance the obvious generalization ability to a certain extent. As can be seen from Figure 7, the extraction effect of the deep learning method was significantly better than that of the traditional machine learning algorithm SVM. The traditional machine learning algorithm SVM had a poor recognition effect on the fuzzy floating raft culture area. The main reason is that the fuzzy image features of the floating raft and cage culture areas were bright white, while the traditional machine learning algorithm SVM cannot learn deeper features, thus the floating raft culture area was mistakenly divided into the cage culture area. However, the deep learning method had a stronger feature extraction ability, thus it could correctly identify the floating raft culture area and cage culture area. Among the prediction results of the three deep learning methods, it is obvious that the classification results of our improved U-Net model were better than those of the other two methods. As can be seen from Figure 7e of the prediction results, the prediction of floating raft breeding areas was more complete, and the edges between different floating raft breeding areas were clearer. This also proves that our improved U- Net model can learn more semantic information and location information.

As shown in Table 1, the classification results of each model were analyzed quantitatively. It can be seen that the *F*1 score and MIoU of the proposed improved U-Net model were significantly higher than those of other classification methods. The *F*1 score of the machine learning algorithm SVM classification result was 68.51%, and the Mean Intersection over Union (MIoU) was 58.03%, both lower than the classification accuracy of other deep learning methods. Although the precision of SVM method was the highest at 82.86%, recall was very low at only 63.03%, resulting in the *F*1 score and MIoU being the worst among all the results. Combined with the classification result diagram of SVM, it can be seen that the areas where the sea–land phase intersected and the image features of aquaculture areas were fuzzy were the main areas with a poor classification accuracy of SVM. This may be due to the fact that the machine learning-based method must first perform a water and land separation operation to separate the land area in the image to obtain an area containing only seawater before classifying the image for offshore aquaculture areas, and then classify the area for aquaculture. Therefore, the classification results were affected to a certain extent by the operation of water and land separation. On the other hand, the feature extraction ability of the machine learning algorithm SVM was weaker than the deep learning method and could not extract more advanced features. The spectral features of offshore aquaculture areas were complex, and the image features were more difficult to extract. In summary, the traditional machine learning algorithm SVM model after a high precision value may be benefit from the land and water separation and only needs to pay

attention to the aquaculture of easy points; recall values, *F*1 low scores, and MIoU may be because the model could not handle difficult points of aquaculture.

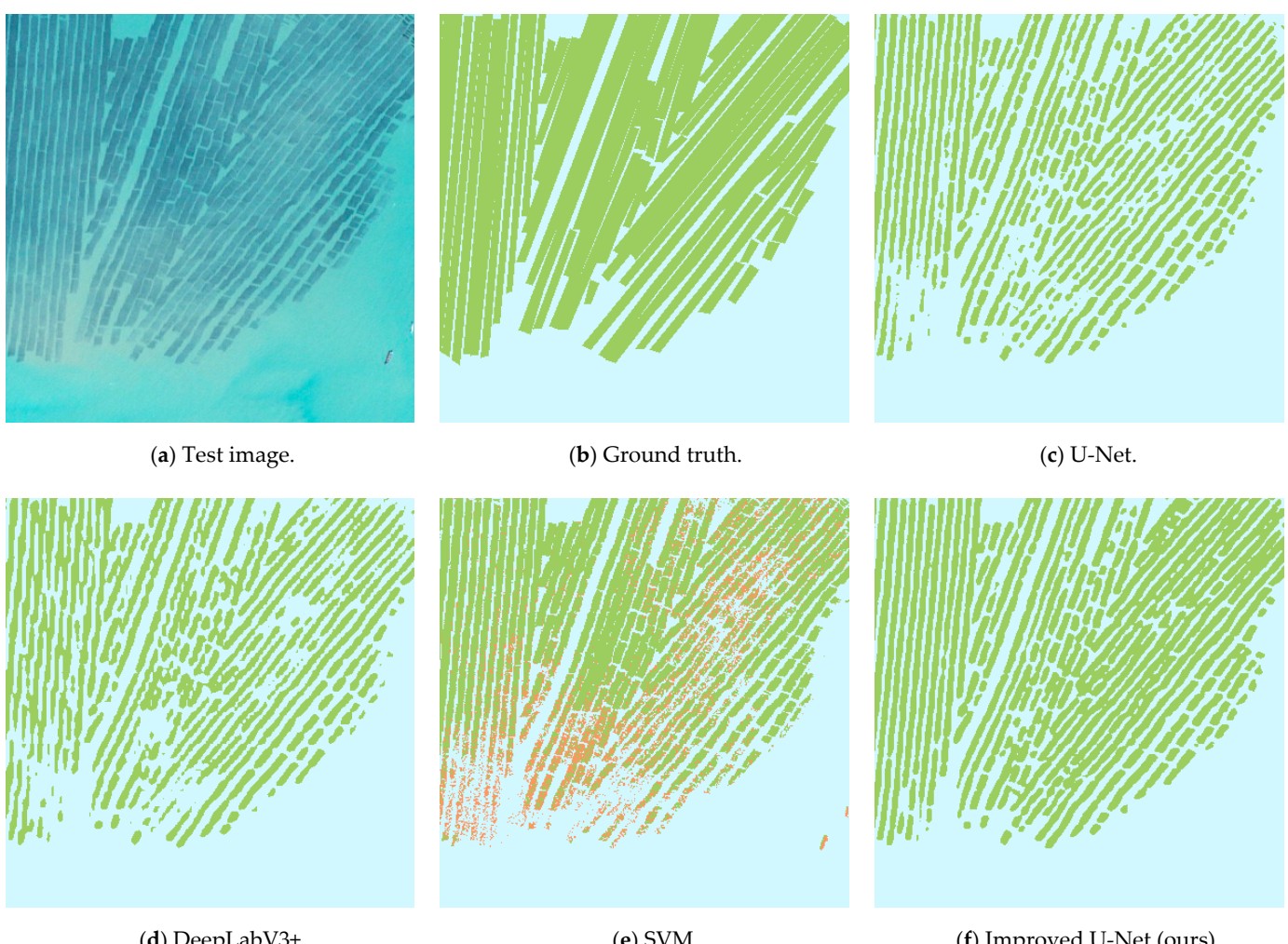

**Figure 7.** Compare the classification results of different models in another typical region. In the figure, the orange area represents the cage culture area; the green area represents the floating raft culture area; and the blue area represents the background.

**Table 1.** Accuracy evaluation table of classification results of different models; the best value is underlined.

| Methods | *Precision* (%) | *Recall* (%) | *F1* (%) | *OA* (%) | Kappa | MIoU (%) |
|---|---|---|---|---|---|---|
| U-Net | 67.92 | <u>90.01</u> | 75.65 | 97.18 | 0.7498 | 64.91 |
| DeepLabV3+ | 70.02 | 88.34 | 77.16 | 97.02 | 0.7339 | 66.04 |
| SVM | <u>82.86</u> | 63.03 | 68.51 | 95.31 | 0.6893 | 58.03 |
| ours | 81.56 | 86.72 | <u>83.75</u> | <u>97.53</u> | <u>0.7924</u> | <u>73.75</u> |

The *F*1 scores and MIoU scores of DeepLabV3+ model were 77.16% and 66.04%, respectively, which were 1.51% and 1.13% higher than those of the original U-Net model. The precision of DeepLabV3+ was slightly higher than that of the original U-Net model, while the recall was slightly lower than that of the original U-Net model. The results show that the original U-NET model and DeepLabV3+ model had their own advantages and disadvantages in the classification of offshore aquaculture. The original U- Net model had a better recall, while the extraction results of DeepLabV3+ model had a better precision. In general, the classification accuracy of the two models was roughly the same.

The MIoU and Kappa coefficients of the improved U-Net model were 83.75%, 73.75%, and 0.7924, respectively. The *F*1 scores, MioU, and Kappa coefficients were also improved by 6.59%, 7.71%, and 0.0426, respectively, compared with the second-best results, as shown in Table 1. The precision of the improved U-Net model was 81.56%, which was better than that of the other methods except the SVM method. In summary, the improved U-Net model had better classification performance compared with common deep learning methods and traditional machine learning algorithms. At the same time, the improved U-Net model had better *F*1 scores and MIoU, which also indicates that the proposed model had better robustness, which is helpful for its application in the extraction task of aquaculture areas in other seas.

*4.4. Ablation Study*

In order to further evaluate the effectiveness of the improved U-Net model proposed by us, ablation experiments were carried out in this section to better verify the effectiveness of the improved ASPP structure and up-sampling structure proposed in this paper All ablation experiments were performed on the same machine, using the same data set for training and validation, and the parameter settings during the training process are also the same. U-Net is the original U-Net model without any improvement. Based on the original U-Net model, U-Net_1 uses the up-sampling structure proposed in this paper instead of interpolation to restore the image size. U-Net_2 is the improved U-Net model proposed in this paper. Compared with U-Net_1, it uses the improved ASPP structure proposed in this paper to connect the encoder part and the decoder part. The results of the ablation analysis are shown in Table 2.

**Table 2.** Accuracy evaluation table of ablative analysis, the best value is underlined.

| Methods | *Precision* (%) | *Recall* (%) | F1 (%) | OA (%) | Kappa | MIoU (%) |
|---------|-----------------|--------------|--------|--------|-------|----------|
| U-Net | 67.92 | 90.01 | 75.65 | 97.18 | 0.7498 | 64.91 |
| U-Net_1 | 73.97 | 89.72 | 80.34 | 97.49 | 0.7861 | 69.91 |
| U-Net_2 | 81.56 | 86.72 | 83.75 | 97.53 | 0.7924 | 73.75 |

The *F*1 score of the U-Net_1 model was 80.34%, and the Mean Intersection over Union (MIoU) was 69.91%. Compared with the original U-Net model, it was increased by 4.69% and 5%, respectively. Except for recall, the accuracy of U-Net model with up-sampling structure is better than that of the original U-Net model. It also shows that the proposed up-sampling structure can better utilize the semantic information and location information of aquaculture regions than the interpolation method.

Compared with the classification results of the U-Net_1 model, the U-Net_2 model with the improved ASPP structure has increased *F*1 score and Mean Intersection over Union (MIoU) by 3.41% and 3.81%, respectively. Compared with U-Net_1, *F*1 score and MIoU are further improved. This shows that the proposed improved ASPP structure contributes to the model to obtain more multi-scale information, which is conducive to the extraction of aquaculture areas.

The above-mentioned ablation experiment results show that the improved ASPP structure and the up-sampling structure composed of the FAM and the ECA module can effectively improve the extraction accuracy of the original U-Net model for offshore floating raft aquaculture and cage aquaculture areas. The combination of the two can further improve the performance of the U-Net model. The improved ASPP can obtain more semantic and location information, while the proposed up-sampling structure can make full use of the acquired information, and the two complement each other. In summary, it is verified that the improved U-Net model proposed in this paper is reliable in extracting medium-resolution remote sensing images from offshore floating raft aquaculture and cage aquaculture areas.

### 4.5. Map Marking

We selected the Sentienl-2 MIS image data of the waters near Sansha Bay and Luoyuan Bay in 2021 (Figure 8). In order to avoid the interference of clouds, the remote sensing images from January to March were used for synthesis. The relevant details of the data are the same as above. Afterward, the improved U-Net model in this study was used to predict the offshore aquaculture area in this area in 2021 based on migration learning prediction, and the distribution map of the aquaculture area in the area was obtained, as shown in Figure 7.

Combined with the original image in 2021, as shown in Figure 7a, the extraction result of the improved U-Net model was the same as the real offshore aquaculture area distribution. The floating raft and the cage aquaculture area were identified, and the coastal land and seawater were also not recognized as floating raft aquaculture or cage aquaculture areas. In summary, the improved U-Net model on medium-resolution remote sensing images can well identify offshore aquaculture areas.

However, in the 2021 distribution map, there were still some cage culture areas that were not identified. At the same time, there was also a certain degree of omission for the relatively fuzzy areas of the aquaculture area on the remote sensing images, and further research is needed in the future.

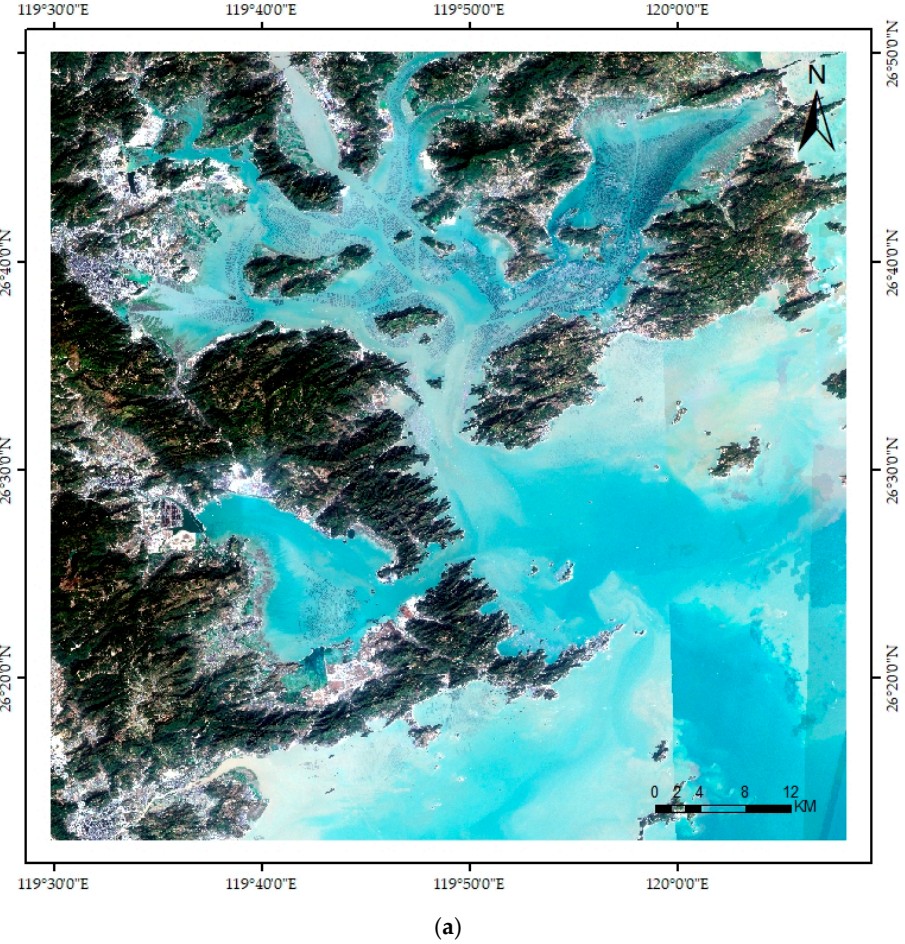

(**a**)

**Figure 8.** *Cont.*

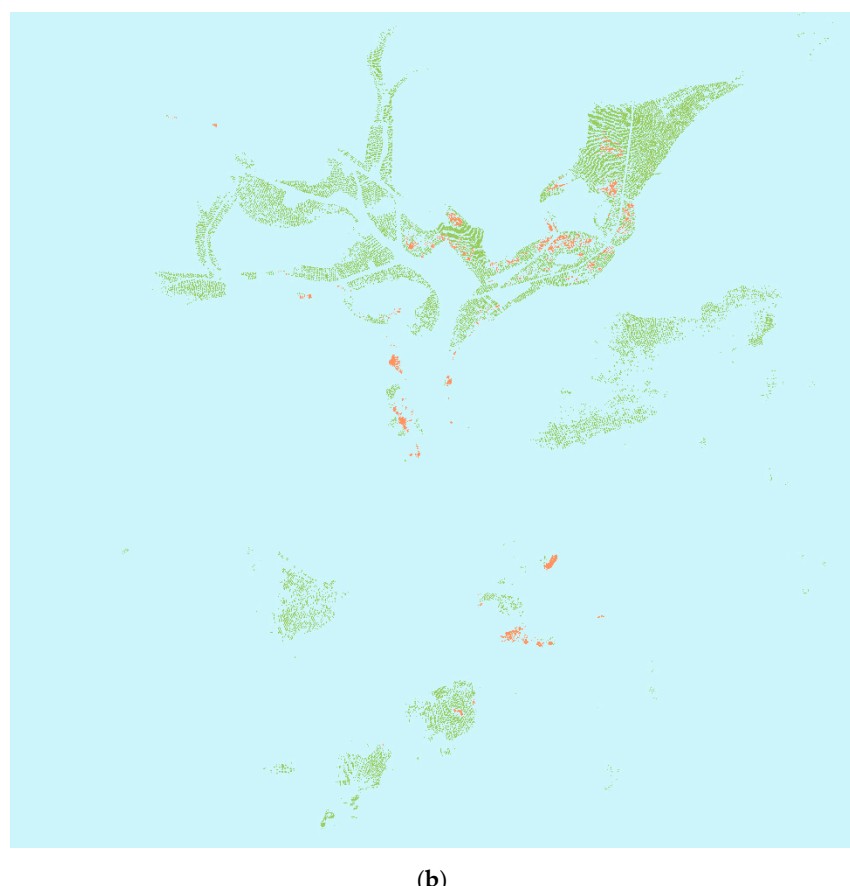

(**b**)

**Figure 8.** The extraction results of aquaculture areas in Sansha Bay and Luoyuan Bay and their adjacent waters in 2021: (**a**) Sentienl_2 MSI images of the study area in 2021; (**b**) the extraction results of aquaculture areas with the improved U-Net model, where blue is the background area including offshore land and sea, green is the extraction result of the floating raft aquaculture area, and orange is the extraction result of the cage aquaculture area.

## 5. Conclusions

We used the Sentinel-2 MSI image data of Sansha Bay and Luoyuan Bay in the northeastern sea area of Fujian Province as the data source. An improved U-Net model suitable for medium-resolution remote sensing image extraction in offshore aquaculture areas is proposed. The model connects the encoder part and the decoder part through an improved ASPP structure and gradually restores the image size of the encoder input feature map through the proposed up-sampling structure. The improved ASPP structure uses strip pooling instead of traditional spatial pooling, allowing the model to obtain multi-scale information while avoiding the redundant information brought by the square window of traditional spatial pooling. At the same time, strip pooling can better identify strip floating raft aquaculture and cage aquaculture areas. The proposed up-sampling structure is composed of a flow alignment module and ECA module, which can make the model better combine different levels of semantic information, make the model pay more attention to the semantic information related to offshore aquaculture area, and effectively avoid the vanishing gradient problem.

We used the improved U-Net model to study the extraction of offshore aquaculture areas from medium-resolution remote sensing images and compared the experimental results with the original U-Net model, the DeepLabV3+ model commonly used in the field of semantic segmentation, and the traditional machine learning algorithm SVM. The experimental results show that the improved U-Net model in this paper was significantly

better than other methods for the extraction performance of offshore aquaculture areas on medium-resolution remote sensing images.

The extraction method proposed in this paper focuses on multi-scale information from image mining, how to make full use of multi-scale information, improving the extraction accuracy and automation degree of offshore aquaculture and providing a basis for relevant departments to conduct large-scale aquaculture monitoring and scientific planning management. It also helps to protect the stability of offshore marine ecosystems and achieve the United Nations Sustainable Development Goal 14 (SDG 14). Finally, due to the relatively low resolution of the remote sensing images, it was more difficult to distinguish the adjacent areas of aquaculture areas, and adhesion and unclear edges between aquaculture areas were more likely to occur. In the future, we will continue to research on our model with the following objectives: (1) to better identify relatively "fuzzy" aquaculture areas in remote sensing images; (2) to further mine edge information from remote sensing images to better identify the boundaries of aquaculture areas; and (3) to further improve the robustness of the model thus that it can be applied to different remote sensing data sources.

**Author Contributions:** Conceptualization, Y.L.; methodology, Y.L. and W.S.; software, W.S.; validation, W.S. and J.S.; formal analysis, W.S. and J.S.; investigation, W.S.; resources, Y.L.; data curation, W.S.; writing—original draft preparation, W.S.; writing—review and editing, Y.L. and J.S.; visualization, W.S.; supervision, Y.L.; project administration, Y.L.; funding acquisition, Y.L. All authors have read and agreed to the published version of the manuscript.

**Funding:** This research was funded by the National Key Research and Development Program of China, grant number 2017YFB0503500, and the Special Projects of the Central Government Guiding Local Science and Technology Development, grant number 2020L3005.

**Data Availability Statement:** The data presented in this work are available on request from the corresponding author. The data are not publicly available due to other ongoing studies.

**Acknowledgments:** The authors would like to thank the editors and reviewers for their detailed comments and efforts toward improving our study.

**Conflicts of Interest:** The authors declare no conflict of interest.

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
