# Peer review of "Extraction of Offshore Aquaculture Areas from Medium-Resolution Remote Sensing Images Based on Deep Learning"

_remotesensing, doi:10.3390/rs13193854_

Round 1

Reviewer 1 Report

The authors present an improved U-Net  model to extract offshore aquaculture areas from medium-resolution remote sensing images based on deep learning. The manuscript is well presented but assumes the reader has  a rather extensive knowledge of deep learning. It would be useful to be convinced of the level of improvement with respect to existing methods a more quantitative treatment in the result section that highlights the significance of the statistics presented. 

Editing suggestions and detail comments below:
Line 71  “, Compared”   ->   “, and compared”
Line 72 “, Which” ->  “, and “
Line 79 “Cui et al.” -> “Cui et al.[18]
Line 116 “MSI” you need to spell out acronym first time in paper
Line 277 “formula” -> “formulae”
Line 307 “yellow” -> “orange”
LIne 338-382 Why is Precision %  not discussed? SVM (Fig 6e) seems to be more similar to the ground truth than the improved U-Net (Fig. 6f) for the cage aquaculture surrounding the island north of the center of the test image. Above what difference are the numbers in table 1 for the different model significantly different?
Line 474 “from the following aspects” -> “with the following objectives”
Line 476 “further” -> “To further”
Line 477 “, it can” -> “, so that it can

Reviewer 2 Report

The paper Extraction of offshore aquaculture areas from medium-resolution remote sensing images based on deep learning by Lu eat al. Deals with a) the improvement of a deep learning framework and b) the detection of different types of aquaculture in Chinese seas.

The paper was interesting to read and I congratulate the authors to their interesting work. I do think the English needs a careful editing, there are a lot of grammar mistakes, wrong capitalizations. Partly, content seems to not have been accurately proof read, for example in lines 210ff and 220ff content seems to be repeating. I would ask the authors to carefully go through the text to make it more readable.

I have a few additional things I suggest to be addressed:

  1. Line 88 ff: the u net description and reference are missing in the introduction. Please state it here, also to better understand your improvement of the model.

  2. lines 447ff : please give an overview and image of the training and validation data? I assume the experiment was run on the Independent 4000x4000 Image. What is the threshold set you used for deciding if there is aquaculture or not?

  3. I wonder whether the OA should be used in this paper (line 282 and throughout the text) given that the large majority of the area is not aquaculture and all models capture tha. the OA values seem way inflated and unrealistic. I would suggest to focus on the other scores, particularly F1.

  4. line 382. please describe what an ablation experiment is first.

  5. line 446. I understand the data is not directly available. But you should provide the code for your improved Unet model either as supplement or a GitHub — how is anybody to repeat, use, or improve on your results otherwise? Also, I suggest to include a magnified image of the fuzzy examples you mention.

again, thank you for our work.

Reviewer 3 Report

< Specific comments>

The manuscript “Extraction of offshore aquaculture areas from medium-resolution remote sensing images based on deep learning” is within the scope of the Journal of Remote Sensing.

In this paper, authors an improved U-Net model especially including an atrous spatial pyramid pooling (ASPP) module and an up-sampling structure is proposed for offshore aquaculture area extraction in this paper. However, it is important for aquaculture monitoring, scientific planning and management to extract offshore aquaculture areas from medium resolution remote sensing images.  Simultaneously, this issue is a very increasingly important issue in fisheries management and Sustainable development goals (SDGs). In the manuscript, the data and methods are thoughtful and relatively well presented. But, some content needed a minor adjustment. Moreover, there are several key problems the author still needs to address.

1. Abstract: The author did not provide enough information in the abstract. Especially, the research goals, research gaps, or that was insufficient technology in past research??

2. Introduction:

(1) Include in the introduction the current/existing debate in the field/topic, hence, why the paper is warranted; and the knowledge gap that the paper is trying to fill. Such statements will establish the reason behind the research and its significance.

(2) Include in the introduction (and later link this in the conclusion) the significance of the study in literature and research of satellite ocean remote sensing. How would the evidence presented in the manuscript be an important contribution to literature and would how can they have potential use for other similar study types or study areas?

3. Introduction and discussion:

This journal has an international audience. How do readers outside China view these results? How do these translate to other world oceans?

4. References: Are all the references and literature citations complete, accurate, and consistent with journal style? Please check.
